# Kibble–Zurek Scaling from Linear Response Theory

**DOI:** 10.3390/e24050666

**Published:** 2022-05-10

**Authors:** Pierre Nazé, Marcus V. S. Bonança, Sebastian Deffner

**Affiliations:** 1Instituto de Física ‘Gleb Wataghin’, Universidade Estadual de Campinas, Campinas 13083-859, Brazil; p.naze@ifi.unicamp.br (P.N.); mbonanca@ifi.unicamp.br (M.V.S.B.); 2Department of Physics, University of Maryland, Baltimore County, Baltimore, MD 21250, USA

**Keywords:** Kibble–Zurek mechanism, linear response theory, quantum phase transition

## Abstract

While quantum phase transitions share many characteristics with thermodynamic phase transitions, they are also markedly different as they occur at zero temperature. Hence, it is not immediately clear whether tools and frameworks that capture the properties of thermodynamic phase transitions also apply in the quantum case. Concerning the crossing of thermodynamic critical points and describing its non-equilibrium dynamics, the Kibble–Zurek mechanism and linear response theory have been demonstrated to be among the very successful approaches. In the present work, we show that these two approaches are also consistent in the description of quantum phase transitions, and that linear response theory can even inform arguments of the Kibble–Zurek mechanism. In particular, we show that the relaxation time provided by linear response theory gives a rigorous argument for why to identify the “gap” as a relaxation rate, and we verify that the excess work computed from linear response theory exhibits Kibble–Zurek scaling.

## 1. Introduction

In thermodynamics a phase transition describes the dramatic change of the macroscopically observable physical properties of matter [1]. At the microscopic scale, such a transition requires the fundamental re-ordering and structuring (or lack thereof) of the system’s constituents. Realizing the complexity of the microscopic properties of a system approaching and passing through a phase transition, it is almost obvious to recognize that around the transition the response to external perturbations is strongly inhibited. In renormalization group theory, this insight is formalized as the universal behavior of response functions [2].

All real processes occur at a finite time and are accompanied by the inevitable production of *nonequilibrium excitations*. If the rate of driving is much slower than the inverse of the relaxation time, effectively quasistatic, equilibrium processes can be facilitated. However, close to critical points the relaxation time diverges (as does the response), and hence any real driving through a phase transition will always exhibit nonequilibrium characteristics. This observation is at the core of the Kibble–Zurek mechanism [3,4,5,6,7,8,9,10,11,12,13,14,15,16,17], which predicts the size of finite domains to be fully determined by the critical exponents and the rate of driving.

Whereas the arguments of the Kibble–Zurek mechanism can be phrased rather intuitively for thermodynamic phase transitions, the situation is more involved for *quantum phase transitions* [18]. A quantum system undergoes a quantum phase transition, if its macroscopically observable physical properties of the *ground state* change according to an external field [18]. It has then been argued that the energy difference between the ground and excited state, the so-called “gap”, plays the role of a relaxation rate, and thus, the Kibble–Zurek mechanism can be generalized to the fully quantum domain [19,20,21,22].

Both, the classical and the quantum Kibble–Zurek mechanism describe nonequilibrium excitations in terms of the critical exponents of the underlying *equilibrium* phase transition. Hence, it appears somewhat natural to assume that the mechanism itself is valid “close enough to equilibrium”. However, like all phenomenological approaches the range of validity cannot be fully determined from within the approach. On the other hand, “close to equilibrium” is the domain of linear response theory [23,24]. Therefore, the natural question arises whether the Kibble–Zurek mechanics can be phrased as a consequence of linear response, or whether the mechanism goes beyond the theory. In previous work, we have found some clear evidence that the Kibble–Zurek mechanism does in fact describe the physics outside the range of validity of linear response [25,26], but also that for slow enough driving, the two approaches are consistent [27].

In the present work, we further investigate to what extent insight from and about the Kibble–Zurek mechanism can be extracted from linear response theory. To this end, we focus on the less intuitive case and analyze the quantum phase transition of the Ising model in the transverse field [28]. Since this model can be solved analytically [29,30], it has become the paradigmatic case study for phase transitions in quantum systems [19,31,32,33,34,35,36]. As a first result, we elucidate the interpretation of the “gap” as a relaxation rate. To this end, we compute the relaxation time directly from the response function, and we find that the quantum phase transition indeed exhibits “critical slowing down”. This insight can then be used to compute the excess work, which quantifies the “amount” of diabatic excitations and which can be computed relatively easily by means of linear response theory [37,38,39,40,41,42,43,44,45,46]. We find that this excess work exhibits exactly the polynomial behavior as a function of the driving time predicted by the Kibble–Zurek mechanism. Finally, benchmarking our results from linear response theory against exact numerics, we obtain a good characterization of the range of validity of linear response theory around quantum phase transitions.

## 2. Preliminaries

We begin by establishing notions and notations. To this end, we briefly review some elements of the Kibble–Zurek mechanism, as well as how to compute the excess work from linear response theory. For specificity, we phrase our analysis entirely in terms of the quantum Ising chain in the transverse field,
(1)$$H=-J\sum_{i=1}^{N}\sigma_{i}^{x}\sigma_{i+1}^{x}-\Gamma \sum_{i=1}^{N}\sigma_{i}^{z}.$$
where $\sigma_{I}^{z}$ and $\sigma_{i}^{x}$ are the Pauli matrices of the *i*th spin, *J* is the coupling energy, and Γ is the transverse magnetic field. For our purposes we choose *N* to be even, and we work with periodic boundary condition.

### 2.1. Kibble–Zurek Mechanism

The Kibble–Zurek mechanism is a phenomenological theory that can be used to describe the non-equilibrium dynamics of the Ising chain (Equation 1) when crossing its critical point, Γ=J. Renormalization group theory predicts [2,18] that the “relaxation time” diverges polynomially governed by the corresponding critical exponents. In quantum phase transitions the energy gap, Δ, plays the role of the relaxation rate [19], and we can write
(2)$$\tau_{R}\left(t\right)\equiv \frac{\hbar }{\Delta (t)},$$
where $\tau_{R}$ is the relaxation time. For large systems, N≫1 it is a simple exercise to show that
(3)Δ(t)≡2|J−Γ(t)|.

For simplicity and without loss of generality [47] we now assume that the magnetic field Γ changes linearly as a function of time,
(4)$$\Gamma \left(t\right)=J\left|1-\frac{t}{\tau }\right|\;,$$
where τ is the duration of the process. The resulting $\tau_{R}\left(t\right)$ is illustrated in Figure 1. Zurek recognized that this “critical freezing out” of the response has crucial ramifications for the nonequilibrium behavior [4]. Far from the critical point, the relaxation dynamics are fast and all nonequlibrium excitations can be mitigated or “healed”. Close to the critical point, this is no longer possible and the nonequilibrium shattering of the order parameter is imprinted onto the system. Thus, the regions far from the critical point are called *adiabatic* and close to the critical point the system undergoes the *impulse* regime.

The transition from adiabatic to impulse behavior occurs when the relaxation time becomes equal to the driving time $\hat{t}=\tau_{R}\left(\hat{t}\right)$, which can be solved for $\pm \hat{t}$. We have
(5)$$\hat{t}=\pm \sqrt{\frac{\hbar \tau }{2J}},$$
which is governed by the driving rate 1/τ, with which the system crosses the critical point.

### 2.2. Excess work in Linear Response Theory

In the following, we will investigate how much of the Kibble–Zurek mechanism is encoded in linear response theory. To this end, it will be instructive to write the Hamiltonian (Equation 1) as
(6)$$H\left(t\right)=H_{0}+A\;\lambda \left(t\right)\;,$$
where *A* is some “observable” and |λ(t)|≪1. We will be particularly interested in the excess work $W_{\mathrm{ex}}$, i.e., the amount of energy above the ground state that is injected due to the finite time driving. In linear response theory $W_{\mathrm{ex}}$ can be written as [25,26,27,37,38,39,40,41,42,45,46,48]
(7)$$W_{\mathrm{ex}}=\frac{1}{2}\int_{0}^{\tau }\int_{0}^{\tau }dt^{\prime }dt\;\Psi \left(t-t^{\prime }\right)\dot{\lambda }\left(t^{\prime }\right)\dot{\lambda }\left(t\right).$$
where $\Psi (t-t^{\prime })$ is the relaxation function. See Ref. [44] for a brief review on $W_{\mathrm{ex}}$ and linear response theory. This can be determined from the response function,
(8)$$\phi \left(t\right)=\frac{1}{i\hbar }{\langle [A(0),A(t)]\rangle }_{0},$$
and $\phi \left(t\right)=-\dot{\Psi }\left(t\right)$. The average is taken over an initial, equilibrium state, here over the ground state wave function, and A(t) is evolved according to the Heisenberg equation of motion for the unperturbed Hamiltonian $H_{0}$.

### 2.3. Excess Work from Kibble–Zurek Arguments

In Ref. [27] it was argued that the behavior of $W_{\mathrm{ex}}$ can be predicted with arguments from the Kibble–Zurek mechanism. To this end, it is instructive to recognize that only driving in the impulse regime will appreciably contribute to $W_{\mathrm{ex}}$, and hence the integrals in Equation (Equation 7) are evaluated up to $\hat{t}$ and not τ. Note that strictly speaking, Ref. [27] verified the claim only for *thermodynamic* phase transitions, and more specifically noise-induced phase transitions. That similar arguments hold for *quantum* phase transitions is at best a sophisticated guess.

However, if one simply works with the expression of the relaxation function from renormalization group theory for the quantum Ising model, it is easy to show that [27]
(9)$$W_{\mathrm{ex}}∼\tau^{\gamma_{\mathrm{KZ}}},\;\gamma_{\mathrm{KZ}}=\frac{\Lambda -2}{z\nu +1\prime }$$
where Λ is the critical exponent corresponding to the variation of an external parameter, *z* the dynamical critical exponent and ν the spatial critical exponent. In the present case, the driven Ising chain, we have Λ=0 for the magnetic fields, z=1, and ν=1, and hence $\gamma_{\mathrm{KZ}}=-1$, which is consistent with numerical findings [25,32]. However, the question remains whether this is a coincidence or a deep conceptual fact. Quantum phase transitions occur in the ground state and in unitary dynamics. Hence, notions such as “relaxation” are borrowed at best, and must not be taken too literally. Hence, a more thorough analysis of the relaxation function for the quantum Ising chain appears instrumental to elucidate how the Kibble–Zurek mechanism arises from the equilibrium properties of isolated quantum systems.

## 3. The Relaxation Function

We now need to analyze the relaxation function, Ψ(t), more thoroughly and determine the corresponding relaxation time (within the framework of linear response theory). In Appendix A, we show that Ψ(t) for the quantum Ising chain (Equation 1) can be written as
(10)$$\Psi \left(t\right)=\frac{16}{J}\sum_{n=1}^{N/2}\frac{J^{3}}{\epsilon_{n}^{3}}\;\sin^{2}\left(\frac{(2n-1)\pi }{N}\right)\cos\left(\frac{2\epsilon_{n}t}{\hbar }\right)\;,$$
where we have introduced the eigenenergies
(11)$$\epsilon_{n}=2\sqrt{J^{2}+\Gamma_{0}^{2}-2J\Gamma_{0}\cos\left(\frac{(2n-1)\pi }{N}\right)}\;,$$
and $\Gamma_{0}\equiv \Gamma \left(t=0\right)$. Observe that Ψ(t) is a highly oscillatory function, which is expected for an isolated quantum system evolving under unitary dynamics. Moreover, the expression describing the relaxation behavior is governed by the initial value of the transverse magnetic field, which is a consequence of linear response theory. Thus, already at this point we recognize that the Kibble–Zurek mechanism goes beyond linear response theory, as its arguments address the simultaneous response of the system to the external driving. We will see shortly in Section 4.1 that this does not lead to a major complication within the range of validity of linear response theory.

### 3.1. Large N Limit

Phase transitions and their corresponding singularities are observed strictly only for infinitely large systems N≫1. In this limit, the discrete eigenvalue spectrum (Equation 11) becomes continuous and the quantum numbers can be expressed in terms of the wavenumber k=2πn/N. Thus, we write,
(12)$$\psi \left(t\right)≃\frac{8J^{2}}{\pi }\int_{0}^{\pi }dk\;\frac{\sin^{2}\left(k\right)}{\epsilon^{3}\left(k\right)}\cos\left(\frac{2\epsilon (k)t}{\hbar }\right)$$
and the eigenenergies (Equation 11) become
(13)$$\epsilon \left(k\right)=2\sqrt{J^{2}+\Gamma_{0}^{2}-2J\Gamma_{0}\cos\left(k\right)}\;.$$

Note that the ground state n=0 now corresponds to the zero mode, k=0.

#### Ferromagnetic and Paramagnetic Phases

It is instructive to first inspect the relaxation function far from the critical point. For $\Gamma_{0}/J\ll 1$ the quantum Ising model (Equation 1) assumes ferromagnetic ordering. In this case, the relaxation function (Equation 12) can be expanded and the leading order is,
(14)$$\psi_{F}\left(t\right)=\frac{1}{2J}\;\cos\left(\frac{4Jt}{\hbar }\right)\;.$$

Such a relaxation function is characteristic for single spins, which is a good description of macroscopic spin ordering. Moreover, observe that this ferromagnetic relaxation function is independent of the external field $\Gamma_{0}$

In the opposite limit, $\Gamma_{0}/J\gg 1$, the Ising chain becomes paramagnetic. The corresponding expansion of Ψ(t) gives in leading order
(15)$$\psi_{P}\left(t\right)=\frac{J^{2}}{2\Gamma_{0}^{3}}\;\cos\left(\frac{4\Gamma_{0}t}{\hbar }\right)\;,$$
which expresses the fact that paramagnetic systems are highly susceptible to external fields. The stark contrast in the response of the ferromagnetic and paramagnetic phases to external driving is indicative of the “dramatic” change that occurs at the phase transition.

#### Divergence at the Critical Point

It is then easy to see that Equation (Equation 12) exhibits a critical divergence if the Ising chain (Equation 1) is driven through its phase transition at Γ=J. To this end, we introduce the amplitude density A(k) as well as the characteristic frequency Ω(k), with which we can write
(16)$$\psi \left(t\right)=\int_{0}^{\pi }A\left(k\right)\cos\left(\Omega (k)t\right)\;.$$

Now assuming that the chain starts close to the critical point, $\Gamma_{0}\approx J$, we obtain
(17)$$A\left(k\right)=\frac{\sin^{2}k}{2\sqrt{2}J\pi {\left(1-\cos k\right)}^{3/2}}$$
and
(18)$$\Omega \left(k\right)=\frac{4\sqrt{2}}{\hbar }\;\left(1-\cos k\right)\;,$$
for which Ψ(t) clearly diverges in the limit k→0. Moreover, note that in this limit $\cos\left(\Omega (k)t\right)$ becomes a constant as a function of time, which is the characteristic “freezing” of the response around the critical point.

#### Variance of the Magnetic Moment Per Spin

For time-independent problems, and for quasistatic driving the relaxation function becomes identical to the magnetic susceptibility, χ, [24]. Thus, we now evaluate χ=Ψ(t=0) for systems prepared in the zero mode, k=0 by directly integrating Equation (Equation 16). We obtain,
(19)$$\chi =\frac{\left(\Gamma_{0}^{2}+J\right)K\left(\frac{4J\Gamma_{0}}{{\left(\Gamma_{0}+J\right)}^{2}}\right)-{\left(\Gamma_{0}+J\right)}^{2}E\left(\frac{4J\Gamma_{0}}{{\left(\Gamma_{0}+J\right)}^{2}}\right)}{\pi \Gamma_{0}^{2}\left|\Gamma_{0}+J\right|},$$
where *K* and *E* are the complete elliptic integral of first and second kind [49]. Equation (Equation 19) is depicted in Figure 2. We observe that, as expected, at the critical point χ diverges, and that decays polynomially into the ferro- and paramagnetic phases.

This establishes that the relaxation function (Equation 12) in the limit N≫1 exhibits important properties of a thermodynamic system undergoing a phase transition. Next, we will show how a corresponding relaxation time can be derived from Ψ(t).

## 3.2. Relaxation Time

In linear response theory, the relaxation time, $\tau_{R}$, can be determined directly from the relaxation function [24]. We have
(20)$$\tau_{R}=\int_{0}^{\infty }dt\;\frac{\Psi (t)}{\Psi (0)}\;,$$
which we can now evaluate for the quantum Ising chain with Equation (Equation 12). Note, however, that for isolated quantum systems the relaxation function (Equation 12) is oscillatory, and hence Equation (Equation 20) is an indeterminate integral. Therefore, in Appendix B, we compute the upper envelop of the integral in Equation (Equation 20), for which we obtain
(21)$$\tau_{R}^{\mathrm{UB}}=\frac{\hbar {\left(J+\Gamma_{0}\right)}^{2}\left(J^{2}+\Gamma_{0}^{2}\right)}{8J^{2}\Gamma_{0}^{2}}\;\frac{1}{|J-\Gamma_{0}|}\;.$$

Equation (Equation 21) is plotted in Figure 3, which closely resembles Figure 1.

Remarkably, the relaxation time determined by means of linear response in Equation (Equation 21) is governed by the gap and we can write
(22)$$\tau_{R}^{\mathrm{UB}}∼{\left|J-\Gamma_{0}\right|}^{-1}\;.$$ Consequently, the critical exponent ν=1, and more importantly $\tau_{R}^{\mathrm{UB}}$, gives a more transparent justification for the identification of the energy gap with a relaxation rate (Equation 2). Equation (Equation 21) constitutes our first main result. Rather than having to rely on plausibility arguments, the relaxation time in isolated quantum systems can be determined directly from the relaxation function of linear response theory.

It appears plausible that this finding holds generally for any many-body system exhibiting a quantum phase transition. However, a more sophisticated analysis or at least a numerical verification may be required to explore whether the inverse gap is generally related to the relaxation time from linear response theory.

## 4. Kibble–Zurek Scaling of the Excess Work

Now that we have established that both relaxation function as well as the corresponding relaxation time behave properly, it is tempting to directly compute the excess work (Equation 7). However, to guarantee that our comparison with predictions from the Kibble–Zurek mechanism are sound, we first need to more carefully analyze the range of validity of linear response theory around the critical point.

### 4.1. Range of Validity

To this end, we computed the exact excess work by solving the corresponding time-dependent Schrödinger equation using a standard Runge–Kutta method. The excess work can be written as
(23)$$W_{\mathrm{ex}}=\langle \psi \left(\tau \right)|H\left(\tau \right)|\psi \left(\tau \right)\rangle -\langle \psi \left(0\right)|H\left(0\right)|\psi \left(0\right)\rangle -\Delta E$$
where ΔE is the exergy [27,50], which reduces to the energy difference of initial and final groundstates. Expressions for ΔE can be found in the literature [51]. The numerically exact results can then be compared with the expression from linear response theory (Equation 7) for the relaxation function in Equation (Equation 10). In Figure 4, we plot our findings for a range of system sizes and “perturbation strengths”, and for processes starting in the ferromagnetic, $\Gamma_{0}>1$, as well as the paramagnetic, $\Gamma_{0}<1$, phases.

Intuitively, we would expect linear response theory to be accurate as long as the quantum Ising chain remains close to its ground state. Thus, a natural parameter to quantify the range of validity can be chosen to be $\delta \Gamma /\epsilon_{1}\ll 1$, where δΓ denotes the “strength” of the driving and $\epsilon_{1}$ is the energy difference between ground and excited state, i.e., the gap. Note that $\epsilon_{1}\to 0$ for N→∞. Thus, one would expect a failure of linear response theory for large systems, which means in the limit of “proper” phase transitions. In fact, in Figure 4, we observe very good agreement between the prediction of linear response theory and the exact numerics for small enough $\delta \Gamma /\epsilon_{1}$. However, we also observe that for large $\delta \Gamma /\epsilon_{1}$ linear response theory still qualitatively captures the behavior of the excess work as a function of the external driving.

Note that the critical point is only crossed in Figure 4g–i. However, also for such processes we find regimes in which linear response theory accurately predicts the excess work, and in all other cases we have at least qualitatively accurate results. Thus, we can now continue to analyze the scaling behavior of $W_{\mathrm{ex}}$ (Equation 7).

### 4.2. Kibble–Zurek Scaling from Linear Response Theory

Based on our understanding for when linear response theory is accurate, we can now verify the expected Kibble–Zurek scaling. To this end, we consider a case of $N=10^{5}$ and a process that drives through the critical point at a constant rate (Equation 4). In complete analogy to Ref. [27] we consider only the excess work accumulated in the impulse regime,
(24)WexIm=J2τ2∫−t^t^∫−t^tdt′dtΨ(t−t′). Note that for each τ we have a corresponding value of $\hat{t}$ (Equation 5), and that we choose $\Gamma_{0}=\Gamma \left(-\hat{t}\right)$. This is a fair analysis as the Kibble–Zurek arguments only depend on the rate of driving, and not on the initial values of the external field. The resulting values of WexIm are plotted on a log-log scale as a function of the driving time τ in Figure 5. We observe polynomial behavior over three orders of magnitude, and the numerical Kibble–Zurek exponent $\gamma_{\mathrm{KZ}}\approx -1$. This is in full agreement with the aforementioned expectation, and we are now comfortable to conclude that the framework developed in Ref. [27] indeed also applies to quantum phase transitions.

## 5. Concluding Remarks

In the present analysis, we analyzed the consistency and interplay of two phenomenological frameworks to describe quantum phase transitions, namely the Kibble–Zurek mechanism and linear response theory. We found that while the Kibble–Zurek mechanism does go beyond the range of validity of linear response theory, additional insight can be obtained by studying both frameworks. A key finding of our analysis is that the relaxation time determined from linear response theory gives solid and rigorous justification for the plausibility argument that identifies the “gap” as a relaxation rate. Moreover, we found that the excess work computed from linear response theory exhibits the scaling properties that are predicted by the Kibble–Zurek arguments.

## Figures and Tables

**Figure 1 entropy-24-00666-f001:**
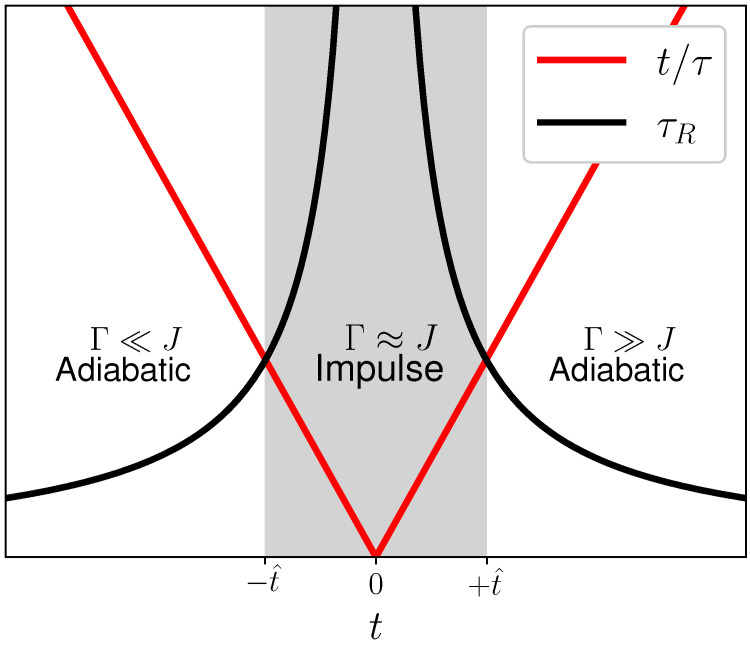
Illustration of the Kibble–Zurek mechanism. Far from the critical point, the dynamics of the system is essentially adiabatic, meaning that the system recovers from the defects of the driving faster than the inverse of the driving rate. Close to the critical point the situation changes dramatically. The healing capacity is lost and finite-size domains are “frozen” into the system.

**Figure 2 entropy-24-00666-f002:**
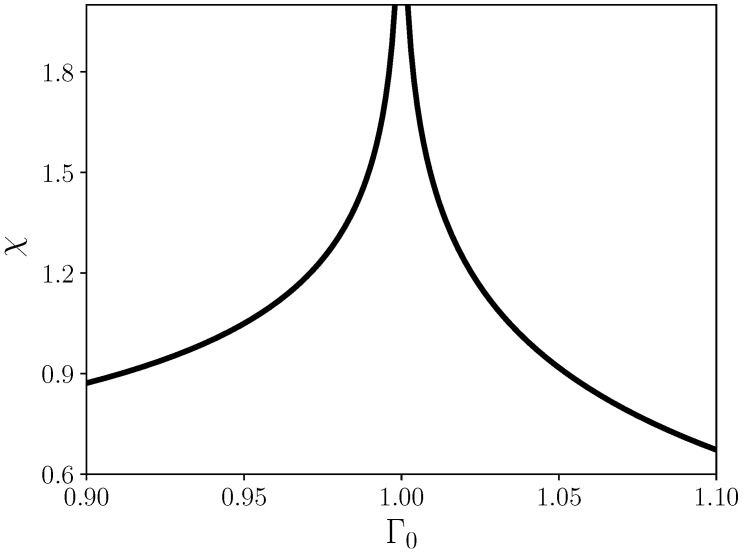
Magnetic susceptibility (Equation 19) as a function of the external field $\Gamma_{0}$ for J=1.

**Figure 3 entropy-24-00666-f003:**
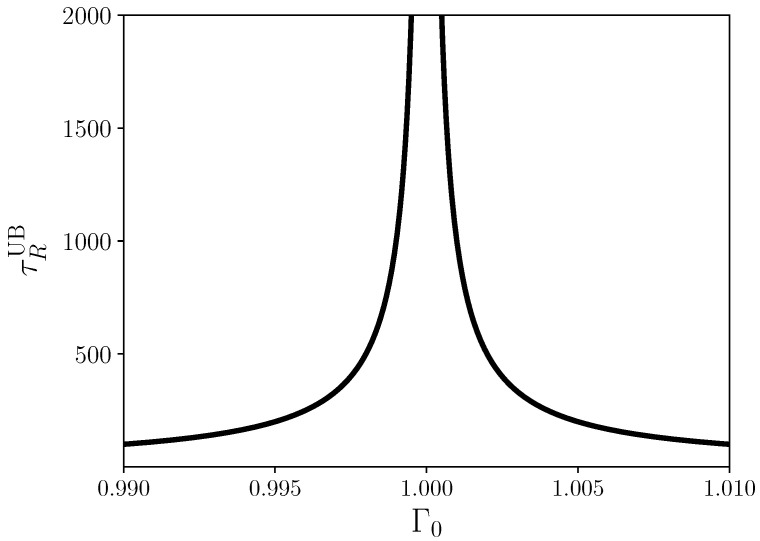
Effective relaxation time (Equation 21) for J=1.

**Figure 4 entropy-24-00666-f004:**
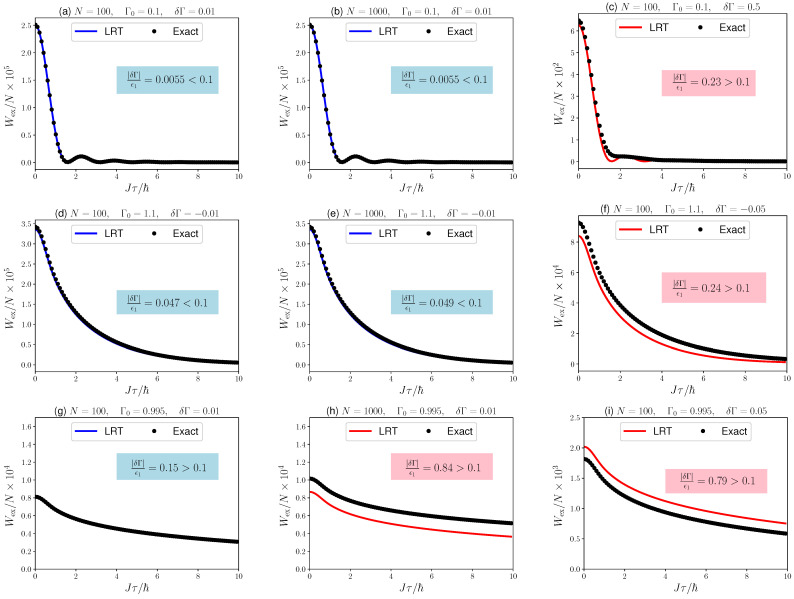
Excess work (Equation 7) computed from linear response theory and exact numerics for protocols driving in the ferromagnetic (**a**–**c**) and paramagnetic (**d**–**f**) phase, and crossing the critical point (**g**–**i**). Figures (**a**,**d**,**g**) depict situations in which linear response theory and the exact result perfectly match. Figures (**b**,**e**,**h**) depict situations with large *N*. Figures (**c**,**f**,**i**) depict situations with strong driving.

**Figure 5 entropy-24-00666-f005:**
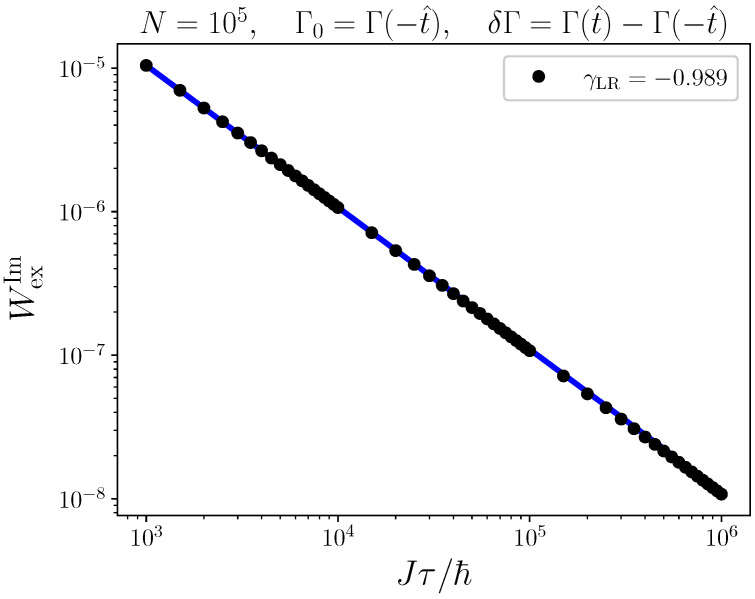
Comparison between Kibble–Zurek scaling of the excess work (Equation 7) from exact dynamics and linear response theory.

## Data Availability

Not applicable.

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
