# Peer review of "Kibble–Zurek Scaling from Linear Response Theory"

_entropy, 2022, doi:10.3390/e24050666_

Round 1
Reviewer 1 Report
The authors present an interesting analysis of Kibble-Zurek mechanism for the quantum phase transition occurring in the quantum Ising model in transverse field.
The paper is clear and well writen and the results interesting. In particular it is interesting that linear response theory predicts the correct KZ exponent for the excess work.
Without hesitation I recommend this work for publication in Entropy.
Below are a couple of remarks that the authors might want to look at before publishing.
1. Regarding Eqs. (7,8). As far as I remember from Kubo's original paper [
R. Kubo, "Statistical-mechanical theory of irreversible processes. I", J. Phys. Soc. Jpn. 12 570--586 (1957)], the relaxation function, Eq. (8) should be expressed [in the quantum regime] in terms of an imaginary-time integral; something of the form:
\int_0^\beta du <A(-i \hbar u) A(t)>
Most likely I am missing something, but why is there no imaginary time integral in Eq. (8)? See also [M. Campisi and S. Denisov and P. H\"anggi
Phys. Rev. A 86, 032114 (2012)] where the dissipated work is expressed as in Eq. (7), but with the relaxation function featuring the imaginary time integration.
2. In regard to the excess work in crossing a quantum phase transition, and its relation to the response function, perhaps it is worth mentioning the work of [L. Fusco et al. Phys. Rev. X 4 031029 (2014)].
3. In the intro, there is a hint that response function always diverge at a critical point. That however is not necessarily always the case, typically the response function goes as a power law, and depending on the sign of the exponent it either diverges or vanishes. E.g.: The measured critical exponent \alpha for the lambda transition of superfluid He, is negative, meaning
C ∼ [(T-T_C)/T_C]^(-\alpha)
goes to zero at the crit. point. Please consider improving the text.
4. Please specify the definition of \Gamma_0 below Eq. (11)
5. Typos:
Line 55. extend ->extent
Fig.4 Caption. situationsin ->situations in. math ->match
line 172. orders of magnitude -> decades
line 255. Physical Review E -> Phys. Rev. E (check that all the refs are abbreviated)
Author Response
See attached pdf

Reviewer 2 Report
The paper investigates the universal dynamics near a quantum phase transition with linear response theory and Kibble-Zurek mechanism. They found the Kibble-Zurek mechanism does go beyond the range of validity of linear response theory near the critical point, while shows well consistent when far from the critical point. The main results of the work show that the relaxation time determined from linear response theory gives solid and rigorous justification for the plausibility argument that identifies the “gap” as a relaxation rate. Moreover, they found that the excess work computed from linear response theory exhibits the scaling properties that are predicted by the Kibble-Zurek arguments. This paper presents rigorous analysis of relaxation time, and exhibits the scaling properties of the excess work predicted by the Kibble-Zurek mechanism. I would like to recommend this paper for publication in this journal if the authors may address the following comments.
- What is the definition of in Eq. (11) ?
- The main results of this work show that the relaxation time determined from linear response theory gives solid and rigorous justification for the plausibility argument that identifies the “gap” as a relaxation rate. Whether it is a universal result for other physical systems.
- We know the KZ critical exponents and can be extracted from systematic excitation gap or equilibrium state properties in the frame of the KZM. Whether the KZ critical exponents and can also be extracted from the linear response theory.
Author Response
See attached pdf

Reviewer 3 Report
In this manuscript, the authors investigate numerically the theory of the Kibble-Zurek mechanism in a quantum system by means of linear response theory. They discuss the critical slowing down and compute the excess work which is found to follow the scaling characteristic of classical scenarios.
The methods used and described in the manuscript are sounds and will be accessible to researchers. I feel the letter will be of interest to the journal readership.
However, I believe the presentation of the work needs to be substantially improved, in order to make this work accessible and informative to the reader.
- Line 20: In the introduction, I believe there is a missing part describing in general terms recent advances in the field of KZM in classical and quantum systems. As an example, the following works should, at least, be cited and used:
-- Keesling, Alexander, et al. "Quantum Kibble–Zurek mechanism and critical dynamics on a programmable Rydberg simulator." Nature 568.7751 (2019): 207-211.
-- Deutschländer, Sven, et al. "Kibble–Zurek mechanism in colloidal monolayers." Proceedings of the National Academy of Sciences 112.22 (2015): 6925-6930.
-- Zamora, A., et al. "Kibble-zurek mechanism in driven dissipative systems crossing a nonequilibrium phase transition." Physical Review Letters 125.9 (2020): 095301.
-- Biroli, Giulio, Leticia F. Cugliandolo, and Alberto Sicilia. "Kibble-Zurek mechanism and infinitely slow annealing through critical points." Physical Review E 81.5 (2010): 050101.
-- Cui, Jin-Ming, et al. "Experimentally testing quantum critical dynamics beyond the Kibble–Zurek mechanism." Communications Physics 3.1 (2020): 1-7.
- Line 69. It would be beneficial to give a better introduction to the concepts of excess work and linear response theory when introduced.
- Line 73. It would be beneficial to give a better description/justification on why the gap plays the role of the relaxation rate.
- Lines 105-112. I suggest rephrasing this paragraph in order to give to the reader a better introduction to the next points.
- Line 122. Here the authors should clarify what is the reasoning behind the introduction of these terms.
- Line 126. The description can be improved by explaining how one arrives to Eq.(19).
- Line 146. More information on the numerical procedure used to obtain the results discussed should be provided.
- To simplify the reading, Figs 2 and 3 could be gathered in the same plot.
- Fig 4 is currently hard to read. I would suggest plotting and discussing more curves in the same inset, keeping two parameters constant while one is varied.
In this way, more cases can be analyzed and discussed.
- Following the above, it would be interesting to understand what are the points where the LRT fails, as a function of the different variables.
- In fig.4 the caption should give a better description of the results presented, without limiting to a bare description of the different parameters tuned.
- Finite-size effects are crucial in critical dynamics. Can the author discuss how the excess work behaves once the system is decreased in size.
- Line 176. In the concluding remarks, it would be instructive to learn how the presented analysis can contribute to future theoretical or experimental works.
Round 2
Reviewer 2 Report
The authors have revised their manuscript according to my report. I'm happy to recommend this version for publication.